# Segment Anything Meets Universal Adversarial Perturbation

## Abstract

As Segment Anything Model (SAM) becomes a popular foundation model in computer vision, its adversarial robustness has become a concern that cannot be ignored. This works investigates whether it is possible to attack SAM with image-agnostic Universal Adversarial Perturbation (UAP). In other words, we seek a single perturbation that can fool the SAM to predict invalid masks for most (if not all) images. We demonstrate convetional image-centric attack framework is effective for image-independent attacks but fails for universal adversarial attack. To this end, we propose a novel perturbation-centric framework that results in a UAP generation method based on self-supervised contrastive learning (CL), where the UAP is set to the anchor sample and the positive sample is augmented from the UAP. The representations of negative samples are obtained from the image encoder in advance and saved in a memory bank. The effectiveness of our proposed CL-based UAP generation method is validated by both quantitative and qualitative results. On top of the ablation study to understand various components in our proposed method, we shed light on the roles of positive and negative samples in making the generated UAP effective for attacking SAM.

## 1 Introduction

With an increasingly important role in driving groundbreaking innovations in AI, deep learning has gradually transitioned from training models for specific tasks to a general-purpose foundation model Bommasani et al. (2021). For language foundation models like BERT Devlin et al. (2018) and GPT Radford et al. (2018; 2019), have made significant breakthroughs in the natural language processing (NLP) area and contributed to the development of various generative AI Zhang et al. (2023a), including the text generation (ChatGPT Zhang et al. (2023b)), text-to-image Zhang et al. (2023c) and text-to-speech Zhang et al. (2023d), text-to-3D Li et al. (2023), etc. On top of the early successful attempts like masked encoder Zhang et al. (2022a), Meta research team has recently proposed a vision foundation model called Segment Anything Model (SAM) Kirillov et al. (2023), which mimics GPT to control the output with prompts. Such a prompt-guided approach alleviates the need for finetuning and thus has impressive zero-shot transfer performance.

After the release of *Segment Anything* project, SAM has been widely used in various applications, such as image editing Kevmo (2023) and object tracking Adamdad (2023); Chen (2023), etc. Therefore, it is critical to understand its robustness in various contexts. Early works Qiao et al. (2023) have examined its generalization capabilities beyond natural images to medical images Zhang et al. (2023e) and camouflaged images Tang et al. (2023). Follow-up works have further evaluated its robustness under style transfer, common corruptions, patch occlusion and adversarial perturbation. Attack-SAM is a pioneering work to study how to attack SAM with adversarial examples, but it mainly focuses on image-independent attacks Zhang et al. (2023f). In other words, the generated perturbation can only be used for attacking the model for a specific image, which requires generating a new perturbation when the image changes. By contrast, a universal adversarial attack seeks a single perturbation (termed UAP) that causes the adversarial effect to all images and leads to wrong label predictions for most images Moosavi-Dezfooli et al. (2017a) in the context of image classification. With the image-agnostic property, the UAP can be generated beforehand and applied to any image for the attack purpose and thus is relatively more practical but also more challenging. Therefore, our work is devoted to studying whether it is possible to attack SAM with a UAP.

Classical adversarial attack methods like DeepFool Moosavi-Dezfooli et al. (2016) and PGD Madry et al. (2018) optimize the perturbation to make the output of the adversarial image different from that of the original clean image. The classical UAP algorithm introduced in Moosavi-Dezfooli et al. (2017a) is based on DeepFool and thus follows such an image-centric approach. This requires access to the original training data and thus FFF Mopuri et al. (2017a) studies PGD-based approaches for generating data-free UAP Mopuri et al. (2017a) with a relatively weaker attack performance. Prior works Qiao et al. (2023); Zhang et al. (2023f) show that such an image-centric approach is also effective for attacking SAM, but the investigation is limited to image-independent attacks. A major difference in generating UAP lies in changing the to-be-attacked training image in every iteration to avoid over-fitting to any specific image. We follow this practice to extend Attack-SAM from image-independent attacks to universal attacks, however, such a preliminary investigation leads to unsatisfactory performance. This is attributed to the change of optimization target from one image to another in the image-centric approach. To this end, this work proposes a new perturbation-centric attack method, by shifting the goal from directly attacking images to seeking augmentation-invariant property of UAP. Specifically, we optimize the UAP in the CL method where the UAP is chosen as the anchor sample. The positive sample is chosen by augmenting the anchor sample, while random natural images are chosen as the negative samples.

For the proposed CL-based UAP generation method, we experiment with various forms of augmentations to generate a positive sample and find that augmenting the UAP by adding natural images yields the most effective UAP for universal adversarial attack. Beyond quantitative verification, we also show visualize the attack performance of the generated UAP under both both point and box prompts. We have an intriguing observation that the predicted mask gets invalid under both types of prompts: getting smaller under point prompts and getting larger under box prompts. Moreover, we present a discussion to shed light on why our generated UAP is effective by analyzing different pairs of inputs for the encoded feature representations. It helps us understand the roles of positive samples and negative samples in our CL-based UAP method for crafting an effective UAP to attack SAM.

## 2 RELATED WORKS

**Segment Anything Model (SAM).** SAM is a recent advancement in the field of computer vision that has garnered significant attention Ma and Wang (2023); Zhang et al. (2023e); Tang et al. (2023); Han et al. (2023); Shen et al. (2023); Kang et al. (2022). Unlike traditional deep learning recognition models focusing solely on label prediction, SAM performs mask prediction tasks using prompts. This innovative approach allows SAM to generate object masks for a wide range of objects, showcasing its remarkable zero-shot transition performance. Researchers have explored the reliability of SAM by investigating its susceptibility to adversarial attacks and manipulating label predictions. Furthermore, SAM has been extensively utilized in various applications, including medical imaging Ma and Wang (2023); Zhang et al. (2023e), and camouflaged object detection Tang et al. (2023). It has also been combined with other models and techniques to enhance its utility, such as combining with Grounding DINO for text-based object detection and segmentation IDEA-Research (2023) and integrating with BLIP or CLIP for label prediction Chen et al. (2023); Park (2023); Li et al. (2022); Radford et al. (2021). SAM has found applications in image editing Rombach et al. (2022), inpainting Yu et al. (2023), and object tracking in videos Yang et al. (2023); Zxyang (2023). More recently, MobileSAM Zhang et al. (2023g), which is significantly smaller and faster than the original SAM, realizes lightweight SAM on mobile devices by decoupled knowledge distillation. With the advent of MobileSAM, it is expected more and more SAM-related applications will emerge, especially in the computation-constrained edge devices. This yields a need to understand how SAM works, for which zhang Zhang et al. (2023h) performs a pioneering study and shows that SAM is biased towards texture rather than shape. Moreover, multiple works Qiao et al. (2023); Zhang et al. (2023i) have shown that SAM is vulnerable to the attack of adversarial examples. Our work also investigates the adversarial robustness of SAM, but differentiates by focusing on universal adversarial attack.

**Universal Adversarial Attack.** Universal adversarial perturbation (UAP) has been first introduced in Moosavi-Dezfooli et al. (2017a) to fool the deep classification model by making wrong label predictions for most images. Unlike the vanilla universal attack by the projected algorithm to generate the perturbations, the SV-UAP Khrulkov and Oseledets (2018) adopts singular vectors to craft UAPs, where the method is data-efficient with only 64 images used to iteratively craft the perturbations. Inspired by the Generative Adversarial Networks (GAN), NAG Mopuri et al.

(2018a) and GAP Perolat et al. (2018) focus on obtaining the distribution of UAPs. To compute the UAPs, these approaches use a subset of the training dataset, however, the attacker might be limited in accessing the training data. Therefore, multiple works explore data-free to generate UAPs. FFF Mopuri et al. (2017b) is pioneering to propose a data-independent approach to generate the UAPs, adopting fooling the features learned at multiple layers. GD-UAP Mopuri et al. (2018b) can generate universal perturbations and transfer to multiple vision tasks. Class-discriminative UAP has been investigated in Zhang et al. (2020a); Benz et al. (2020) to fool the model for a subsect of classes while minimizing the adversarial effect on other classes of images. They opt to train the UAP with Adam Optimizer Kingma and Ba (2015) instead of adopting sign-based PGD algorithms Goodfellow et al. (2015); Madry et al. (2018), and such a practice has also been adopted in Zhang et al. (2020b; 2021). In contrast to prior works adopting image-centric DeepFool or PGD to optimize the UAP, our work proposes a perturbation-centric framework with a new UAP generation method based on contrastive learning.

**Self-supervised Contrastive Learning (CL).** With the goal of learning augmentation-invariant representation, for which CL is a miltstone development of unsupervised learning Schroff et al. (2015); Wang and Gupta (2015); Sohn (2016); Misra et al. (2016); Federici et al. (2020). CL consists of positive pair and negative pairs. Unlike the negative pairs, the positive pair are obtained from the same image but differ in augmentation to ensure they have similar semantic information. Earlu works on CL have adopted margin-based contrastive losses Hadsell et al. (2006); Wang and Gupta (2015); Hermans et al. (2017), and NCE-like lossWu et al. (2018); Oord et al. (2018) has later emerged to become the de facto standard loss in CL. For example, classical CL methods like SimCLR Chen et al. (2020a) and MoCo families He et al. (2020); Chen et al. (2020b) adopt the InfoNCE loss which combines mutual information and NCE. Specifically, it maximizes the mutual information between the representation of different views in the same scene.

## 3 BACKGROUND AND PROBLEM FORMULATION

### 3.1 PROMPT-GUIDED IMAGE SEGMENTATION

Segment Anything Model (SAM) consists of three components: an image encoder, a prompt encoder, and a lightweight mask decoder. The image encoder adopts the MAE He et al. (2022) pre-trained Vision Transformer (ViT), which generates the image representation in the latent space. The prompt encoder utilizes positional embeddings to represent the prompt (like points and boxes). The decoder takes the outputs of image and prompt encoders as the inputs and predicts a valid mask to segment the object of interest. In contrast to classical semantic segmentation performing pixel-wise label prediction, the SAM generates a label-free mask. With $x$ and $p$ denoting the image and prompt, respectively, we formalize the mask prediction of SAM as follows:

$$y = SAM(x, p; \theta), \tag{1}$$

where $\theta$ represents the parameter of SAM. Given a image $x \in \mathbb{R}^{H \times W \times C}$, the shape of y is $\mathbb{R}^{H \times W}$. We set the $x_{ij}$ as the pixel values at the image x with the coordinates i and j. $x_{ij}$ belongs to the masked area if the pixel value $y_{ij}$ is larger than the threshold of zero.

### 3.2 UNIVERSAL ADVERSARIAL ATTACK ON SAM

Here, we formalize the task of universal adversarial attack on SAM. Let $\mu$ denote the distribution of images in $\mathbb{R}^{H \times W \times C}$. In the image recognition tasks, the adversary goal is to fool the model to predict wrong labels. Universal adversarial attack, under the assumption that the predicted labels of clean images are the correct ones, seeks a *single* perturbation vector $v \in \mathbb{R}^{H \times W \times C}$ termed UAP to cause label changes for *most* images Moosavi-Dezfooli et al. (2017a). In other words, it aims to maximize the adversarial effect of the UAP in terms of the fooling rate, the ratio of images whose predicted label changes after adding the UAP Moosavi-Dezfooli et al. (2017a). In the context of SAM, the predicted outputs are masks instead of labels and thus the attack goal is to cause mask changes. We follow Attack-SAM to adopt the widely used Intersection over Union (IoU) in image segmentation to evaluate such mask changes. The mIoU calculates the mean IoU for $N$ pairs of clean mask $Mask_{clean}$ and adversarial mask $Mask_{adv}$ shown in Equation 2.

$$mIoU = \frac{1}{N} \sum_{n=1}^{N} IoU(Mask_{clean}^{(n)}, Mask_{adv}^{(n)}), \tag{2}$$

where all the adversarial masks $Mask_{adv}$ are generated for all $N$ images by a single UAP. The goal of universal adversarial attack on SAM is to seek such a single perturbation $v$ to decrease the mIoU defined in Eq. 2 as much as possible. The UAP $v$ is bounded by a $l_p$ norm, which is set to $l_\infty$ norm conventions in prior works on SAM Moosavi-Dezfooli et al. (2017a;b).

**Implementation details.** Considering the image-agnostic property, $N$ in Eq. 2 needs to be larger than 1 and is set to 100 in this work. For the prompts, we randomly choose point prompts unless specified otherwise. Specifically, we randomly select 100 test images from the SA-1B dataset Kirillov et al. (2023) for evaluating the generated UAP. Note that the test images cannot be used for generating the UAP. Following the existing works on the universal adversarial attacks in computer vision, we use $10/255$ as the maximum limit for the perturbation. In other words, the allowed maximum change on each pixel can be no bigger than $10/255$.

## 4 METHOD

### 4.1 EXISTING IMAGE-CENTRIC ATTACK FRAMEWORK

For the task of adversarial attack, the goal is to make the deep model predict invalid output after adding a small perturbation on the input image. Therefore, numerous attack methods, including classical DeepFool Moosavi-Dezfooli et al. (2016) and PGD Madry et al. (2018), optimize such an adversarial perturbation to make the output of adversarial image different from that of its clean image. Such an image-centric approach consists of two steps. First, it predicts the output of clean image $y_{clean}$ and saves it as the ground-truth[1]. Second, the perturbation in the adversarial image is optimized to make $y_{adv}$ different from the ground-truth $y_{clean}$.

Universal adversarial attack requires the perturbation to be effective at random unseen images. Therefore, the to-be-attacked training image needs to be changed in every iteration of the optimization process to avoid over-fitting on any single training image. Such an image-centric approach has been adopted in Zhang et al. (2023i) to demonstrate successful image-independent attacks, and we have adapted it to image-agnostic, *universal* adversarial attacks. The results in Table 1 show that the generated UAP performs much better than random uniform noise sampled between $-10/255$ and $10/255$. Nonetheless, the value of mIoU (59.50%) is still quite high, demonstrating that the UAP is not sufficiently effective for causing mask changes. We also experiment with not changing the to-be-attacked image, which fixes the same optimization goal and results in a successful image-dependent attack with a mIoU of 0.0%. This suggests that a successful attack in SAM requires a consistent optimization target (like attacking a single image). However, such success is limited to image-dependent attacks due to overfitting and cannot be generalized to unseen test images.

Table 1: mIoU (%) results of Image-centric attack by uniform noise and adversarial examples. Image-agnostic indicates the universal setup to attack unseen images.

| Input | Image-dependent | Image-agnostic |
|---|---|---|
| Uniform noise | 86.97 | 86.97 |
| Adversarial attack | 0.0 | 59.50 |

### 4.2 PROPOSED PERTURBATION-CENTRIC ATTACK FRAMEWORK

The above image-centric method is suitable for image-independent attack on SAM but fails for universal attack. The image-centric method is in essence a supervised approach where $y_{clean}$ plays the role of ground-truth and the added perturbation is optimized to make $y_{adv}$ far from $y_{clean}$. Such a supervised approach inevitably causes a dramatic change to the optimization goal when the training

---

[1]the ground-truth output might be given at first in some cases, where this step can be skipped.

image is changed at every iteration. In other words, the failure of image-centric approach for universal attack is conjectured to be the inconsistent optimization goal caused by the change of training image at every iteration. Therefore, we shift the perspective from image to perturbation, which results in our proposed perturbation-centric method. Specifically, in contrast to the predicted masks of the clean and adversarial images, we focus on the independent features of the UAP, which is motivated by perceiving the UAP as an independent input considering its image-agnostic property. How to optimize the UAP in such a perturbation-centric approach, however, is a non-trivial task. It cannot be straightforwardly optimized in a supervised manner as in the image-centric method. To this end, we turn to a widely used self-supervised approach known as *Contrastive Learning (CL)*. The difference between image-centric and perturbation-centric framework is summarized in Figure 1.

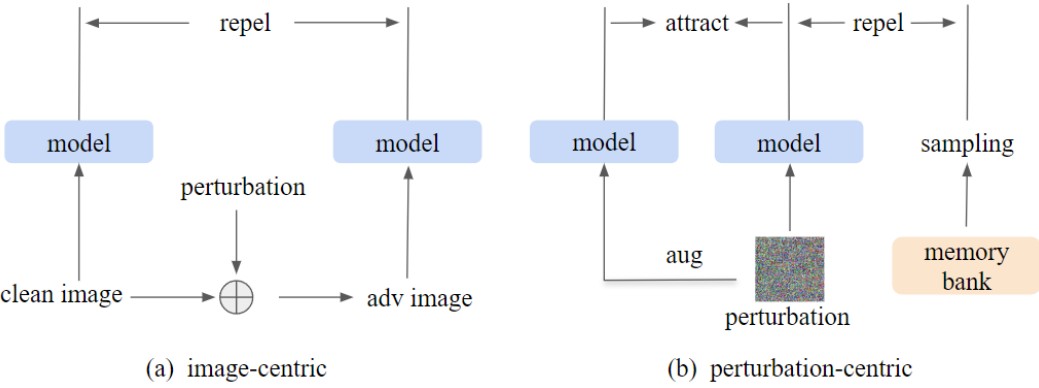

(a) image-centric        (b) perturbation-centric

Figure 1: Difference between image-centric (left) and perturbation-centric (right) attack frameworks.

**CL-based UAP Genration Method.** Outperforming its supervised counterpart, self-supervised learning has become a dominant approach for pre-training an backbone encoder, where CL is a widely adopted method. In the classical CL , there are three types of samples: anchor sample, positive sample, and negative sample. The anchor sample is the sample of interest, while the positive sample is augmented from the anchor sample. Other random images are chosen as the negative samples, and we adopt the same practice in our CL-based UAP generation method. What makes it different from the classical CL method lies in the choice of anchor sample. Specifically, the UAP ($v$) is chosen as the anchor sample because it is the input of interest in this context. For the positive sample, we obtain it by augmenting the anchor sample UAP, which will be discussed in detail. The NCE-like loss (often termed InfoNCE loss) has been independently introduced in multiple works and constitutes as the de-facto standard loss for CL. Following He et al. (2020), we denote the encoded features of the anchor sample, positive sample, and negative sample with $q$, $k_+$ and $k_-$, respectively. Note that the encoded features are often L2 normalized to remove scale ambiguity, based on which the InfoNCE loss adopted in the CL-based UAP generation method is shown as follows:

$$L_{infonce} = -log\frac{exp(q \cdot k_+/\tau)}{exp(q \cdot k_+/\tau) + \sum_{i=1}^{K} exp(q \cdot k_-^i/\tau)}, \tag{3}$$

where $\tau$ represents the temperature controlling the hardness-aware property and thus has an implicit influence on the size of negative samples Wang and Liu (2021); Zhang et al. (2022b). A large negative sample size is required to better sample the high-dimensional visual space He et al. (2020). We follow prior works to save the encoded features of negative samples in a list termed as memory bank Wu et al. (2018) or dictionary He et al. (2020). Since the to-be-attacked SAM encoder does not change during the optimization of UAP, the list does not need to be updated as in classical CL method Wu et al. (2018); He et al. (2020). In other words, the $k^-$ in Eq 3 can be generated once and then saved for sampling during the optimization of UAP.

In the classical CL method, augmentation is applied to ensure augmentation-invariant property for the encoder learning meaningful representations. In our CL-based UAP method, augmentation is also essential for making the generated UAP cause augmentation-invariant feature response on the encoder. This yields two intertwined questions: (1) how should we choose such augmentation for making the UAP effective? (2) why does such augmentation-invariant property makes the UAP effective? The following section performs an empirical study to shed insight on these two intertwined questions.

## 5 EXPERIMENTAL RESULTS AND ANALYSIS

### 5.1 TOWARDS FINDING EFFECTIVE AUGMENTATION

**Preliminary investigation.** In the classical CL method, there are mainly two types of augmentations Chen et al. (2020a). The first type involves spatial transformation like crop/resize and cutout. The second type involves no spatial transformation but causes appearance change by adding low-frequency content (like color shift) or high-frequency content (like noise). We experiment with both types of augmentation and the results are shown in Table 2. We observe that the mIoU values with augmentation crop/resize and cutout consistently remain high, at 85.11% and 75.48%, respectively. It suggests that the spatial transformation is not an effective augmentation type in our UAP generation method. For the second type of adding content, adding uniform noise is also not effective with a mIoU value of 81.14%. By contrast, the augmentation of color shift yields a mIoU of 61.64%, which is comparable to that of the image-centric method (see 59.5% in Table 1).

Table 2: Comparison of different augmentations. The Crop size is 200×200 out of 1024×1024, cutout size is 200×200. The uniform noise and color shift are ranged from 0 to 255. Adding natural images achieves significantly better performance than other augmentations.

| Augmentation type | mIoU ($\downarrow$) |
|---|---|
| Crop/Resize | 85.11 |
| Cutout | 75.48 |
| Uniform noise | 81.14 |
| Color shift | 61.64 |
| Adding natural images | 15.01 |

**From color shift to natural images.** Our preliminary investigation suggests that color shift is the most effective augmentation among those we investigate. We believe that this might be connected to how the generated UAP is applied to attack the model in practice. Since UAP is directly added to the images without spatial transformation, which explains why spatial transformation is less effective. Moreover, natural images have the property of being locally smooth and thus mainly contain low-frequency content, which justifies why the color shift is relatively more effective than adding noise. Motivated by the above interpretations, we conjecture that replacing the color shift images with random natural images for additive augmentation is beneficial for higher attack performance, which is supported by the results in Table 2. Here, for simplicity, the weight of the augmented natural images is set to 1. However, it can be set to values different from 1 (see the ablation study results in Figure 4).

### 5.2 QUALITATIVE RESULTS

It is worth highlighting that our generated UAP has one hidden merit it can generalize to all prompts because the UAP is optimized only on the SAM encoder. In other words, it is truly universal in the sense of being both image-agnostic and prompt-agnostic. In the above, we only report the quantitative results under random point prompts. Here, for qualitative results, we visualize the attack performance under both point prompts and box prompts, with results shown in Figure 2 and Figure 3, respectively. We find that the single UAP causes the model to produce invalid masks for both types of prompts but with an intriguing distinction. Specifically, under the point prompts, the predicted mask region gets smaller with a boundary close to the chosen point prompt. Under the box prompt, however, the predicted mask gets larger than the original mask. We have no clear explanation for this intriguing phenomenon. A possible explanation is that the UAP tends to cause the predicted output to have similar values, *i.e.* causing confusion between the original masked regions and unmasked regions. For the point prompt, the unmasked region tends to be much larger than that of the masked region and thus the predicted mask gets smaller after UAP. By contrast, the box prompts tends to predict a mask inside the box, and thus tends to make the predicted mask boundary get larger and vauge. Note that we can still observe the glass mask in the third row of Figure 3, but the mask boundary gets blurred.

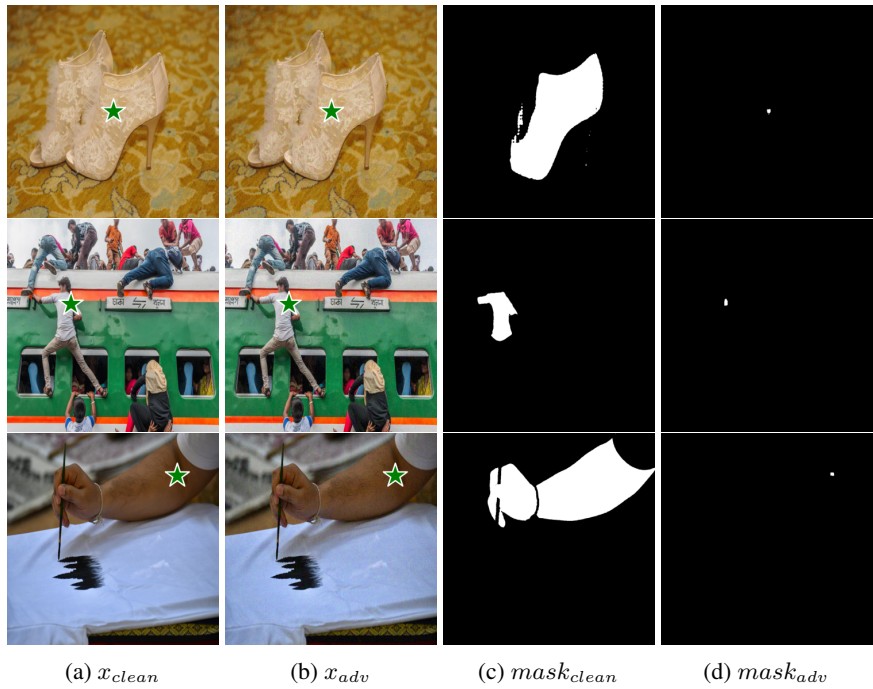

$\quad$ (a) $x_{clean}$ $\qquad$ (b) $x_{adv}$ $\qquad$ (c) $mask_{clean}$ $\qquad$ (d) $mask_{adv}$

Figure 2: Qualitative results under point prompts. Column (a) and (b) shows the clean and adversarial images with the point prompt marked in a green star, with their predicted masks shown in column (c) and (d), respectively. The UAP makes the mask invalid by removing it (or making it smaller).

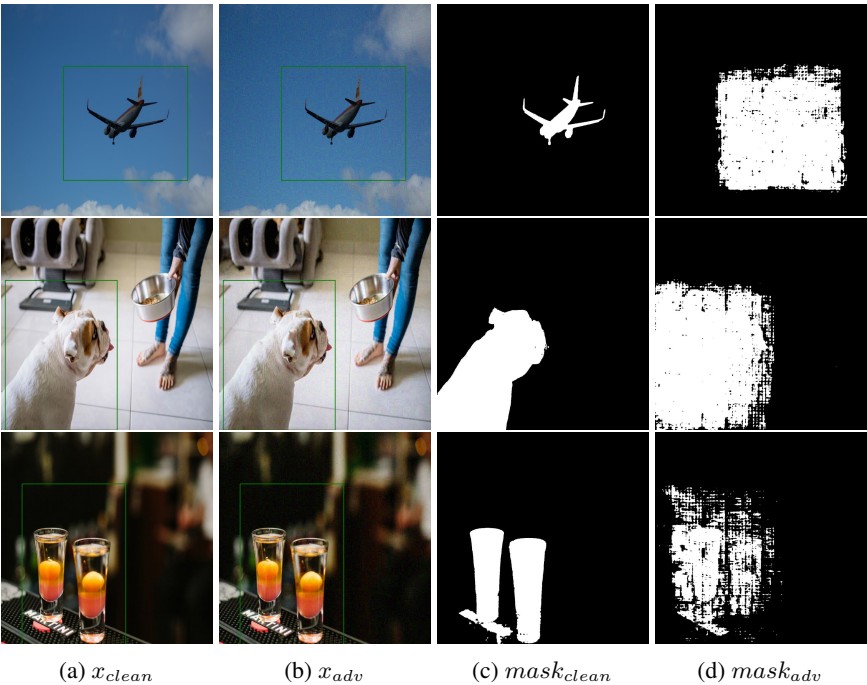

$\quad$ (a) $x_{clean}$ $\qquad$ (b) $x_{adv}$ $\qquad$ (c) $mask_{clean}$ $\qquad$ (d) $mask_{adv}$

Figure 3: Qualitative results under box prompts. Column (a) and (b) refers to the clean and adversarial images with the box prompt marked with green lines, with their predicted masks shown in column (c) and (d), respectively. The UAP makes the mask invalid by making it larger and blurry.

5.3 ABLATION STUDY

**Weight of Augmented Images.** Here, we first conduct an ablation study on the weight of the augmented images. The results are shown in Figure 4. We observe that the mIoU value decreases first increases and then decreases when the weight value is increased from 0.2 to 2 with an interval of 0.1. The strongest attack performance with the mIoU value of 14.21 appears when the weight is set to 1.2. Overall the mIoU value stays low for a relatively wide range of augmentation weight, suggesting our proposed method is moderately sensitive to the choice of augmentaiton weight.

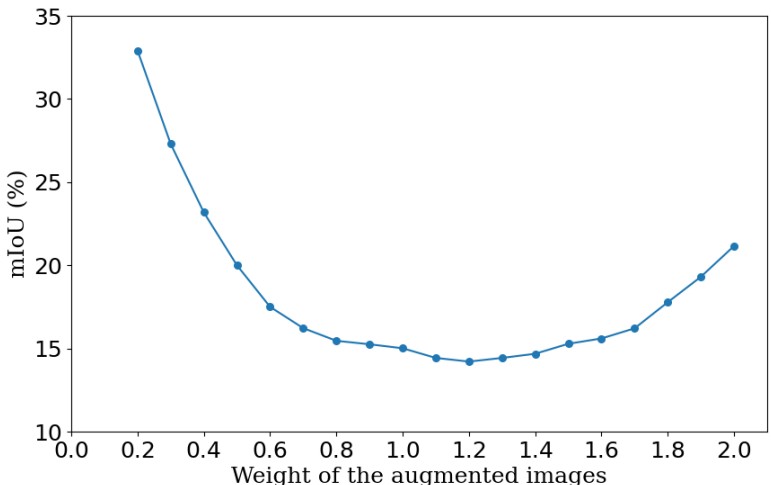

Figure 4: The mIoU (%) results for different weights of the augmented images.

**Size of Negative Sample.** For negative samples in contrastive learning, unlike the positive samples that aim to attract the anchor, our objective is to create a repelling effect on the anchor. This enables the anchor to more effectively foucs on independent features by being drawn towards the positive samples. To accomplish this, it is essential to incorporate a diverse set of negative sample representations, thus avoiding repetitive generation. Therefore, we implement the memory bank mechanism, as do in prior work. We use various sample numbers ( 1, 2, 5, 10, 20, 50, 100) as our memory bank. As shown in Table 3, we observe a significant increase in universal attack performance as the number of samples increases. This indicates that augmenting diverse negative sample representations through the memory bank is beneficial for UAP training. To further augment diverse negative sample representations,

Table 3: The mIoU (%) results on different negative samples $N$.

| N | 1 | 2 | 5 | 10 | 20 | 50 | 100 |
|---|---|---|---|---|---|---|---|
| mIoU ($\downarrow$) | 38.91 | 30.71 | 24.83 | 19.88 | 17.63 | 15.92 | 15.01 |

**Temperature.** Temperature is widely known to have a large influence on the performance of CL method Wang and Liu (2021); Zhang et al. (2022b). The influence of temperature in our CL-based UAP method is shown in Table 4. By default, the temperature is set to 0.1 in this work. We observe that the temperature significantly decreases when the temperature is set to a large value. The reason is that a smaller temperature causes more weight on spent on those hard negative samples Wang and Liu (2021); Zhang et al. (2022b). As revealed in Zhang et al. (2022b), a small temperature is equivalent to choosing a small negative sample size. Therefore, it is well expected that the attack performance decreases when the temperature is set to a sufficiently small value because the a relatively large negative sample size is required for CL. Unlike classical CL, a relatively large temperature does not cause a performance drop.

Table 4: The mIoU (%) results on different InfoNCE temperatures.

| Temperature | 0.005 | 0.01 | 0.05 | 0.1 | 0.5 | 1 |
|---|---|---|---|---|---|---|
| mIoU ($\downarrow$) | 64.61 | 60.58 | 22.78 | 15.01 | 13.28 | 13.48 |

## 5.4 DISCUSSION

To shed more light on why the generated UAP is effective in attacking unseen images, we analyze the cosine similarity of different pairs of inputs for the encoded feature representations, and the results are shown in Table 5. The positive sample pairs have a much higher cosine similarity than that of the negative sample pairs, which aligns with our training objective in Eq 3. The cosine similarity between pairs of adversarial images and its clean images is higher than that of the negative sample pairs, which is excepted because the adversarial image consists of a random natural image and the UAP. The fact that the cosine similarity between positive sample pairs is very high (0.87) suggests that the UAP has independent features and it can be robust against the augmentation of image addition, which aligns with the finding in Zhang et al. (2020b). This partly explains why the cosine similarity between pairs of clean images and adversarial images is relatively low (0.40), causing a successful universal attack. In other words, how the generated UAP attacks the model does not intend to identify the vulnerable spots in the clean images to fool the model as suggested in Moosavi-Dezfooli et al. (2017a;b), but instead form its own augmentation-invariant features.

For the role of negative samples in Eq 3, we find that it is can be at least partially attributed to the the existence of common feature representations regardless of the image inputs for the image encoder, which is supported by a Cosine similarity value of 0.55 higher than zero for pairs of random images. With a list of negative samples in Eq 3, the UAP is expected to be optimized to offset such common features, thus causing adversarial effects. This interpretation is partially supported by the comparison between 0.40 and 0.55 in Table 5.

Overall, the success of Eq 3 for generating an effective UAP can be interpreted as follows: the role of the positive sample is to make the UAP have independent features that are robust against the disturbance of natural image, while the role of negative images facilitates the UAP to find more effective directions to cause adversarial effects by partially canceling out the common feature representations in the image encoder. We leave further detailed analysis to future works.

Table 5: Cosine similarity analysis with different pairs of inputs.

| Input pairs | Cosine similarity |
|---|---|
| Positive sample pairs (UAP and augmented UAP) | 0.87 |
| Negative sample pairs (UAP and random image) | 0.34 |
| Pairs of adversarial image and its clean image | 0.40 |
| Pairs of two random images | 0.55 |

## 6 CONCLUSION

Our work is the first to study how to perform adversarial attack SAM with a single UAP. We demonstrate that existing image-centric attack framework is effective for image-dependent attacks but fails to achieve satisfactory performance for universal adversarial attacks. We propose a perturbation-centric attack framework resulting in a new generation method based on contrastive learning, where the UAP is set to the anchor sample. We experiment with various forms of augmentations and find that augmenting the UAP by adding a natural image yields the most effective UAP among all augmentations we have explored. The effectiveness of our proposed method has been verified with both qualitative and quantitative results. Moreover, we have presented and analyzed different pairs of inputs for the encoded feature representations, which shed light on the roles of positive samples and negative samples in our CL-based UAP method for crafting an effective UAP to attack SAM.

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
