# OpenReview forum: "Segment Anything Meets Universal Adversarial Perturbation"
_ICLR.cc/2024/Conference — ICLR 2024 Conference Withdrawn Submission_

### Official Review · Reviewer_K3ep · 2023-10-31

**Soundness:** 3 good
**Presentation:** 1 poor
**Contribution:** 2 fair
**Rating:** 3
**Confidence:** 4

**Summary:**

This paper focuses on the adversarial robustness of the Segment Anything Model (SAM) in the context of computer vision. The authors investigate the possibility of attacking SAM using a single image-agnostic Universal Adversarial Perturbation (UAP), which can mislead SAM into predicting invalid masks for most, if not all, images. They propose a novel perturbation-centric framework that leverages self-supervised contrastive learning (CL) to generate UAPs effectively.

**Strengths:**

1. The authors are working on a very cutting-edge problem, exploring whether generalized adversarial attacks can be made against segmented large model SAMs.

**Weaknesses:**

1. The authors' approach is not general enough for SAM only in my opinion, especially nowadays there are a lot of variants of SAM and similar generalized segmentation large models like HQ-SAM [1], Semantic-SAM [2], and SEEM [3]. The authors should study attacking segmentation large models, not only SAM.
2. The author didn't quote the reference correctly, all of them are \citet, while some places should be \citep.
3. Multi-modal prompts are not only point and bbox but also text. Both SAM and SEEM [3] can accept text prompts and the author did not address this aspect of the attack.

References

[1] Ke, L., Ye, M., Danelljan, M., Liu, Y., Tai, Y. W., Tang, C. K., & Yu, F. (2023). Segment Anything in High Quality. NeurIPS 2023.

[2] Li, Feng and Zhang, Hao and Sun, Peize and Zou, Xueyan and Liu, Shilong and Yang, Jianwei and Li, Chunyuan and Zhang, Lei and Gao, Jianfeng. Semantic-SAM: Segment and Recognize Anything at Any Granularity. arXiv preprint arXiv:2307.04767.

[3] Xueyan Zou*, Jianwei Yang*, Hao Zhang*, Feng Li*, Linjie Li, Jianfeng Wang, Lijuan Wang, Jianfeng Gao^, Yong Jae Lee. SEEM: Segment Everything Everywhere All at Once. NeurIPS 2023.

**Questions:**

The author's experimental section is very inadequate. Why are there no comparisons with other attack methods and against previous classical segmentation models？

---

> ### Comment · Reviewer_K3ep · 2023-11-22
>
> The author does not give a rebuttal, so my opinion remains unchanged.

---

### Official Review · Reviewer_HSyT · 2023-10-31

**Soundness:** 3 good
**Presentation:** 2 fair
**Contribution:** 2 fair
**Rating:** 5
**Confidence:** 5

**Summary:**

This paper presents a universal adversarial attack against the SAM model. The underlying motivation of this paper is straightforward. However, by simply extending the image-dependent attacks for universal adversarial perturbation, the results are not as good as reported. Instead, this paper presents a new method for the UAP problem against SAM using the contrastive learning perspective. The proposed method is more effective than the baseline method.

**Strengths:**

1.Fortunately, this paper is not simply extending traditional UAP methods to SAM. The newly proposed method based on contrastive learning is more effective than a direct extension of existing methods. This helps a lot in assessing the novelty of the paper.

**Weaknesses:**

1.[minor] There are some typesetting issues in the \cite formats. Please carefully read the template instructions and use \citep or \citet instead. The formatting issue makes the paper difficult to read when printed, especially the paragraphs with dense citations.

2.[minor, defense, ethics] Discussion on how to improve SAM robustness is missing. Although not required, I still want to see some discussions on how we can improve the adversarial robustness of SAM, especially from the unique experience of the proposed UAP. I think there are a couple of references when discussing this. For reference, "On the Robustness of Segment Anything" (https://arxiv.org/pdf/2305.16220.pdf) analyzes the adversarial robustness of SAM. "Enhancing Adversarial Robustness for Deep Metric Learning" (https://openaccess.thecvf.com/content/CVPR2022/html/Zhou_Enhancing_Adversarial_Robustness_for_Deep_Metric_Learning_CVPR_2022_paper.html) presents defense methods for deep metric learning, which is the supervised version of contrastive learning and also involve anchor, positive, negative samples. Moreover, the proposed method resembles "finding a universal visual prompt (perturbation) that can reduce the mIoU". Is it possible to find a "universal visual prompt (perturbation)" that makes SAM more adversarially robust? Those discussions are suggested because, after all, attacks will become robustness evaluation metrics eventually.

3.[important, transferability across prompt type] According to last paragraph in section 3, during evaluation, point prompts are used by default for quantitative evaluations. Is the UAP created under point prompts still effective under other types of SAM prompts? We can never assume the user to stick at one single prompt type. Visualizations for box prompts in section 5.2 without quantitative results are insufficient and not convincing enough at this point. A true "universal" perturbation should not build any correlation with a prompt type. Namely, in the context of SAM, image-agnostic is no longer sufficient for being "universal". It has to be prompt-agnostic as well.

4.[evaluation dataset size] According to the last paragraph in section 3, only 100 images are used for evaluating the proposed method, which does not seem sufficient. UAP evaluation should not be slow as it is merely applying UAP to the image and doing the forward pass. It is suggested to increase the number of test images and additionally report the error bar to make sure the performance is less affected by the bias of the sampled dataset. SA-1B consists of 11M dimages and 1.1B high-quality segmentation masks. The 100 subset is really too small.

5.[clarify, figure] The core formulation of this paper is Eq. 3. To ease reading and understanding, please consider adding the mathematical notations in Figure 1.

6.[important, cost] It would be good to know how much the computational cost is for the proposed method, compared to image-dependent ones. This is because, the higher the attack cost is, the less likely an attacker in practice will adopt it. Hence, attacks with higher costs imply lower practical security risk. This is one of the reasons why I said attacks will eventually serve as robustness metrics. If the authors tend to write the paper from the attack side, then attack cost is important information. If the authors tend to write the paper from the defense side, the robustness discussion should not be absent. The current draft contains neither of them.

**Questions:**

See weaknesses. I'll consider changing my rating based on the author's response on weaknesses marked as "important".

---

### Official Review · Reviewer_8LBE · 2023-11-02

**Soundness:** 2 fair
**Presentation:** 2 fair
**Contribution:** 2 fair
**Rating:** 1
**Confidence:** 5

**Summary:**

This paper explores the problem of creating Universal Adversarial Perturbation (UAPs) for SAM in-order to disrupt its mask prediction ability.

**Strengths:**

The idea of creating UAPs for SAM is interesting.

**Weaknesses:**

- Table 1 is a known phenomenon (Moosavi-Dezfooli et al. (2017a)). I do not see the relevance of this in the paper.

- The idea to increase the strength of the perturbations using contrastive loss has also been explored in works like [A, B]. The only difference I observe is the positive pair chosen in the proposed method.

- With the above point, this also explains the phenomenon of using unrelated natural images yield better results, also observed in [B] (where unrelated natural image patches are compared in the CL).

- The proposed method is severely lacking in terms of comparisons to prior works that attack dense predictions tasks.

[A] GAMA: Generative Adversarial Multi-Object Scene Attacks, NeurIPS 2022

[B] Leveraging Local Patch Differences in Multi-Object Scenes for Generative Adversarial Attacks, WACV 2023

**Questions:**

None.